



# Near-real-time vegetation monitoring and historical database (1981-present) for the Iberian Peninsula and the Balearic Islands

Magí Franquesa[1], Fergus Reig[1], Manuel Arretxea[2], Maria Adell-Michavila[1], Amar Halifa-Marín[1], Daniel Vilas[3], Santiago Beguería[3], Sergio M. Vicente-Serrano[1]

[1]Instituto Pirenaico de Ecología (IPE-CSIC), Consejo Superior de Investigaciones Científicas, Campus de Aula Dei, Zaragoza, E-50059, Spain
[2]Instituto de Geociencias (IGEO), Consejo Superior de Investigaciones Científicas–Universidad Complutense de Madrid (CSIC–UCM), Madrid, Spain
[3]Estación Experimental de Aula Dei (EEAD-CSIC), Consejo Superior de Investigaciones Científicas, Campus de Aula Dei, Zaragoza, E-50059, Spain

*Correspondence to*: Magí Franquesa (magi.franquesa@ipe.csic.es)

**Abstract.** Systematic monitoring and assessment of vegetation dynamics and changes are essential for informing environmental management and conservation strategies. Addressing this need, our study introduces a pioneering procedure to generate a database of vegetation indices that provides semi-monthly updates from 1981 to the present at a 1.1 km spatial resolution, focusing on the Iberian Peninsula and the Balearic Islands. This database enables near-real-time monitoring and analysis of vegetation anomalies. The methodology developed combines harmonized historical satellite imagery from AVHRR, MODIS, and VIIRS sensors. The database's performance was assessed, demonstrating highly accurate and consistent harmonization of NDVI data over time. Notably, the database is adept at identifying temporal variability and trends in vegetation activity and detecting disturbances caused by fire and other phenomena. This work not only advances our understanding of vegetation dynamics in the region but also serves as a crucial tool for policymakers, environmental managers, and agricultural stakeholders. By providing near-real-time updates and using indices to monitor vegetation anomalies, the data allows for comparisons across seasons and vegetation types. The database, which includes the NDVI and kNDVI vegetation indices as well as their standardized versions, SNDVI and SkNDVI, is accessible via https://doi.org/10.20350/digitalCSIC/16201 (Franquesa et al., 2024).

## 1 Introduction

Vegetation indices obtained from remote sensing imagery are used for a wide variety of uses, including the assessment of vegetation dynamics (Bellone et al., 2009; Ferreira and Huete, 2004; Hilker et al., 2014; Malo and Nicholson, 1990; Zhang et al., 2017), long-term vegetation changes (Eckert et al., 2015; Pettorelli et al., 2005) and the assessment of the influence of disturbances like droughts (Breshears et al., 2005; Zeng et al., 2023), wildfires (Cuevas-González et al., 2009; Escuin et al., 2008; Liu et al., 2023) or floods (Powell et al., 2014; Shrestha et al., 2017). Among them, the Normalized Difference



Vegetation Index (NDVI) has long been a cornerstone in remote sensing, offering a robust, quantifiable measure of vegetation health and dynamics. By comparing the difference in near-infrared and red-light reflectance, NDVI provides a crucial insight

into vegetation vigor, biomass, and photosynthetic activity (Myneni et al., 1995; Tucker, 1979). This index's significance is particularly pronounced in the context of climate change, where monitoring vegetation responses is vital for understanding and adapting to environmental shifts (Nemani et al., 2003; Piao et al., 2020).

To this aim, numerous NDVI products at different temporal and spatial resolution have been derived from Earth Observation satellite data, primarily from the Advanced Very High-Resolution (AVHRR) instrument on board National Oceanic and

Atmospheric Administration (NOAA) satellites, and the Moderate Resolution Imaging Spectroradiometer (MODIS) instrument on board Terra and Aqua satellites (Beck et al., 2011; Scheftic et al., 2014; Tucker et al., 2005). The extensive temporal data records provided by AVHRR and MODIS sensors support the creation of comprehensive NDVI time-series that span several decades. For instance, the Global Vegetation Greenness from AVHRR GIMMS-3G+ dataset (Pinzon et al., 2023) provides NDVI data for more than 40 years, from 1981 to 2022, with a spatial resolution of 0.0833 degrees (~ 9.25 km) and a

semi-monthly temporal resolution. Similarly, Li et al. (2023) have recently released the PKU GIMMS NDVI product, an enhanced version of the GIMMS NDVI dataset, covering nearly the same period (from 1992 to 2022) and offering equivalent spatial and temporal resolutions. Additionally, MODIS Terra and Aqua Vegetation Indices (VIs) datasets have been made available operationally, providing NDVI data since early 2000 at different spatial resolution (250 m, 500 m, 1km and 0.05 degrees, equivalent to approximately 5.5 km) (https://modis.gsfc.nasa.gov/data/dataprod/mod13.php, last access in July,

2024). Moreover, with the nearing end of the operational life of the Terra and Aqua satellites, the Visible Infrared Imaging Radiometer Suite (VIIRS) on board the Suomi National Polar-orbiting Partnership (Suomi NPP) and NOAA-20 (JPSS-1) satellites is poised to continue the legacy of MODIS NDVI products (Benedict et al., 2021; Endsley et al., 2023). VIIRS is designed to provide data continuity with similar spatial resolutions and enhanced sensing capabilities. Already, VIIRS NDVI products are being developed to match the quality and resolution of MODIS, ensuring the ongoing availability of high-

resolution vegetation monitoring datasets (https://viirsland.gsfc.nasa.gov/Products/NASA/VIESDR.html, last access in July, 2024).

The coarse spatial resolution (over 9 km for GIMMS NDVI datasets) or the limited temporal coverage of MODIS (since 2000) and VIIRS (since 2012) NDVI datasets restrict their applicability, most notably in regions with highly fragmented and diverse landscapes. In this sense, only a few regions have NDVI products with medium spatial resolution and long-term time series.

Examples include the AVHRR NDVI composites for the conterminous U.S. and Alaska (1989-2019, 1 km resolution), which are no longer generated due to NOAA 19 satellite orbit degradation (Earth Resources Observation and Science Center, 2019), and the corrected representation of the NDVI using historical AVHRR satellite images (1987-2023, 1 km resolution) produced by the Government of Canada for crop monitoring (Government of Canada, 2024). For Europe, a 1 km spatial resolution NDVI product based on AVHRR data has been developed starting from the early 1980s but has not been released to the public (Asam

et al., 2023). For the Iberian Peninsula and the Balearic Islands, the AVHRR NDVI product, Sp_1km_NDVI, (1981 to 2015, 1.1 km resolution) is available but the time-series only extends until 2015 (Vicente-Serrano et al., 2020).

NDVI raw data can effectively show the condition of vegetation at a certain moment, but it does not allow for direct comparison between different locations or across different times. To address this limitation, several standardized vegetation indices (SVIs) such the Vegetation Condition Index (VCI) (Kogan, 1990), the Vegetation Productivity Indicator (VPI) (Sannier et al., 1998) or the Standardized Vegetation Index (SVI) (Peters et al., 2002) have been proposed to derive vegetation condition anomalies. These indices compare current NDVI values relative to statistics extracted from the historical NDVI archive, thereby indicating how much an NDVI value deviates below (negative anomaly) or above (positive anomaly) from 'normal' conditions. Computing robust SVIs requires long-term time series that are not always available, and researchers often must rely on short-term NDVI time series (Anon, 2001; Atkinson et al., 2011; Chiang and Chen, 2011; Li et al., 2014; Nanzad et al., 2019). This reliance on short time series can limit the reliability of the SVI values. In addition, another inconvenience of the SVIs arises from the standardization methodology. For instance, the SVI assumes a normal distribution of the data during normalization, which is not always the case (Peters et al., 2002). This can lead to inaccuracies, complicating spatial and temporal comparative analysis, and thus affecting the reliability of the SVI for monitoring vegetation anomalies.

The lack of long-term continuity and finer spatial NDVI data hinders the effective monitoring and analysis of vegetation dynamics in complex environments, emphasizing the need for comprehensive datasets that bridge these gaps (Vicente-Serrano et al., 2020). Recognizing the mentioned limitations of existing NDVI datasets, underscores the broader challenge of capturing and analyzing complex environmental dynamics across diverse landscapes and the long-term. This challenge is not unique to vegetation monitoring but extends into the broader domain of global change processes, where the need for precise and adaptable tools to assess different environmental processes, including the monitoring of anomalies in vegetation activity, is critical.

Our research fills these critical gaps, presenting the longest continuously updated NDVI time series for the Iberian Peninsula and the Balearic Islands to support both retrospective and immediate vegetation assessments. By meticulously harmonizing AVHRR, MODIS and VIIRS NDVI data, we offer a new dataset that provides a comprehensive historical overview (since 1981 to present) of vegetation dynamics in the region. Moreover, we include in our database the recently developed kernel Normalized Difference Vegetation Index (kNDVI) (Camps-Valls et al., 2021), which is less affected than NDVI by vegetation saturation problems under high leaf area index. Additionally, the dataset features robust standardized versions, SNDVI and SkNDVI, all of which are integrated into a near-real-time vegetation monitoring system. This addition aims to provide a more consistent framework for comparing vegetation anomalies across different regions and time periods, thereby advancing vegetation monitoring in the Iberian Peninsula and the Balearic Islands.

## 2 Data

To build the long-term vegetation indices (VIs) database, we utilized a variety of NDVI products sourced from three distinct satellite sensors: AVHRR, MODIS, and VIIRS. Detailed information regarding the specifications, temporal coverage, and resolution of each dataset is provided as follows:



### 2.1 Sp_1km_NDVI

We used the 1.1 km spatial resolution NDVI dataset for the entire Iberian Peninsula and the Balearic Islands, referred as Sp_1km_NDVI (Vicente-Serrano et al., 2020). This dataset spans three decades, from 1981 to 2015, and is based on daily NOAA–AVHRR afternoon images. Before compositing, the daily AVHRR imagery underwent a series of preparatory steps, such as adjusting and harmonizing NOAA-7, 9, 11, 14, 16, 17 and 19 satellite AVHRR images through cross-calibration, aligning images geographically, removing cloud contamination, deriving reflectance values at the top of the atmosphere, and

adjusting for terrain effects (Poroarson et al., 2021; Vicente-Serrano et al., 2020). Following this preprocessing, the NDVI was computed and composited on a semi-monthly basis using the Maximum Value Compositing (MVC) technique (Holben, 1986), providing two NDVI layers each month. The first layer, labeled as day 1, represents the highest NDVI value observed in the first half of the month, and the second layer, labeled as day 15, represents the highest NDVI value from the second half of the month. The accuracy of the Sp_1km_NDVI dataset was assessed by seasonal and annual comparison with other three widely

used global NDVI products: GIMMS3g, SMN, and MODIS. These comparisons revealed similar spatial and temporal patterns to those observed in the referenced products (Vicente-Serrano et al., 2020). The dataset employs the Universal Transverse Mercator (UTM) projection system, referenced to the European Datum of 1950 (ED50), within Zone 30N. The Sp_1km_NDVI dataset is available for downloading and visualization at http://spainndvi.csic.es/ (last access in July, 2024).

### 2.2 MYD13A2 C61

The Aqua MODIS MYD13A2 Version 6.1 dataset (Didan, 2021) delivers NDVI at a 1km spatial resolution and 16 days temporal resolution. This product selects the highest quality pixel based on minimal cloud cover and optimal viewing angles from all available data within each 16-day period. Featuring alongside the primary vegetation layers are two quality assurance layers, reflectance bands (red, near-infrared, blue, and mid-infrared), and observation layers. The dataset has been validated to Stage 3 according to the validation stages defined by the Land Product Validation (LPV) subgroup (https://lpvs.gsfc.nasa.gov/,

last access in July, 2024), ensuring high reliability with an accuracy of ±0.025 for under ideal conditions. These error estimates are based on rigorous comparisons with ground-based and other satellite data across diverse ecosystems (http://tinyurl.com/zt3uttab, last access in July, 2024). The MYD13A2 V6.1 data is available at https://e4ftl01.cr.usgs.gov/MOLA/MYD13A2.061 or https://lpdaac.usgs.gov/products/myd13a2v061 (last access in July, 2024).

### 2.3 VNP13A2 V002


The VNP13A2 product offers NDVI generated from afternoon observations captured by the VIIRS sensor aboard the NASA/NOAA Suomi NPP satellite. It is produced at a spatial resolution of 1 km and a temporal resolution of 16 days, matching the same specifications of the MODIS-derived MYD13A2 product. Furthermore, VIIRS-based NASA VIs have achieved Stage 1 validation according to the LPV subgroup standards. Preliminary assessments of VIIRS VIs through comparison with





ground-based data collected at AErosol RObotic NETwork (AERONET) sites indicated an overall accuracy of 0.009 for the

Top Of Canopy (TOC) NDVI (Shabanov et al., 2015). This accuracy represents the average deviation of the measured values

from the true or reference values. The VNP13A2 V002 dataset is available at https://lpdaac.usgs.gov/products/vnp13a2v002/

(last access in July, 2024). To simplify reading in the rest of the document, the products Sp_1km_NDVI, MYD13A2 C61, and

VNP13A2 V002 will hereafter be referred to as AVHRR, MODIS, and VIIRS NDVI products, respectively.

## 3 Methods

### 3.1 Data preprocessing

Prior to applying statistical methods for dataset harmonization, essential pre-processing steps were executed to align the spatial

and temporal resolutions of the NDVI products derived from AVHRR, MODIS, and VIIRS sensors. For spatial alignment,

bilinear resampling was used to adjust the 1 km resolution of the MODIS and VIIRS NDVI products to match the 1.1 km

resolution of the AVHRR NDVI product. Temporally, the 16 days resolution of MODIS and VIIRS NDVI products does not

align with the semi-monthly schedule of the AVHRR product. To reconcile these differences and achieve temporal

synchronization, we employed a linear interpolation to estimate NDVI values at the desired intervals. To achieve this, daily

NDVI values were estimated from the MODIS and VIIRS products for the entire period using the 'approx' function in R, and

subsequently, the average of these values was calculated for the biweekly periods corresponding to the AVHRR product.

### 3.2 Statistical data harmonization

The objective of statistical harmonization is to reduce discrepancies among different datasets, thereby creating a consistent,

long-term record. Despite that AVHRR and VIIRS NDVI data overlap for a period of about three years, from 2012 to early

2015, the inclusion of MODIS to bridge AVHRR and VIIRS data provides a longer overlap period with both AVHRR (13

years) and VIIRS (>20 years) (Fig. 1), offering an extended window for cross-calibration and adjustments between the sensors,

which may enhance the accuracy and consistency of the composite time series.

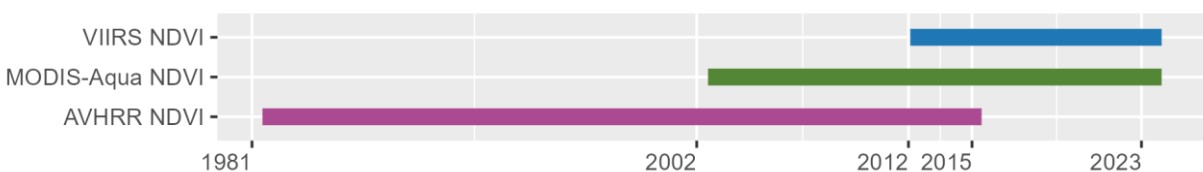

**Figure 1. Temporal cover of the AVHRR, MODIS, and VIIRS NDVI datasets.**

This integration involved a two-step harmonization process using their overlapping periods. Firstly, we harmonized the NDVI

data from the AVHRR-based product (i.e., Sp_1km_NDVI) to align with MODIS-based product (i.e., MYD13A2 C61), and

secondly, we harmonized the AVHRR-MODIS NDVI dataset to align with the VIIRS NDVI data (i.e., VNP13A2 V002). This

approach ensured that all three NDVI datasets were consistent with each other.



To harmonize the NDVI data from one dataset to the other, we tested two different methods: (i) Quantile Normalization (Beguería et al., 2019), and (ii) Ratio-based harmonization. Each method is described in detail below to elucidate how they contribute to reducing discrepancies and aligning the datasets.

**3.2.1 Quantile Normalization**

Quantile Normalization involves calculating the quantiles of the empirical data using probability distributions that best fit the data characteristics of each NDVI product during their overlapping periods. This method consists of several key steps:

(a) Use of different probability distributions: for this purpose, we analyzed independently the 24 semi-monthly series. Each series was standardized by means of an optimal approach that selects the most suitable probability distribution to fit the

data, following standard approaches used to calculate standardized hydrological (Vicente-Serrano et al., 2012) and climatic indices (Moccia et al., 2022; Slavková et al., 2023; Tam et al., 2023). For this purpose, ten probability distributions were tested (i.e., exponential, gamma, generalized extreme value, generalized logistic, generalized pareto, generalized normal, three-parameter lognormal, normal, Pearson type III and Weibull), which are widely used to fit hydroclimate series (e.g., Hamed and Rao, 2019).

(b) Fitting the probability distributions: in order to guarantee an accurate harmonization of the NDVI series, we calculated the parameters of the different probability distributions that fit the semi-monthly series considering the overlapping periods between the different NDVI datasets (2002-2015 between AVHRR and MODIS and 2012-2021 between MODIS and VIIRS). The parameters obtained from each one of the semi-monthly series of the different distributions corresponding to the overlapping periods were calculated using L-moments (Hosking, 1990).

(c) Cumulative Distribution Function (CDF) Calculation: With the estimated parameters, the CDFs for each distribution were calculated following Hosking (1990), transforming the NDVI data to cumulative probabilities. The result of this step is ten different series of cumulative probabilities corresponding to every semi-monthly series for each NDVI series of each pixel and data set (AVHRR, MODIS and VIIRS).

(d) Generation of standardized series: The series of cumulative probabilities obtained by means of the ten probability

distributions were standardized in order to select the best one for each semi-monthly NDVI series for each pixel. Transformation of the cumulative probabilities to z-units was done by means of the method of Abramowitz and Stegun (1965).

(e) Selection of the optimal standardized series: We used the Shapiro-Wilks normality test, which has been used in similar contexts by previous studies to determine the suitability of different distributions (Naresh Kumar et al., 2009; Stagge et

al., 2015). This approach involves assessing the goodness of fit of different probability distributions to fit series of data. The p-values offer valuable insight into the degree of normality of the standardized series; higher p-values approaching 1 indicate greater adherence to a normal distribution so we have chosen the distribution with the highest p-value in the test across the ten series. This approach ensures flexibility in the selection of the probability distribution, based on the genuine statistical characteristics of the series, rather than being constrained by a predetermined theoretical distribution.



(f)   Series harmonization: The cumulative probabilities corresponding from the NDVI series of the non-overlapping period between AVHRR and MODIS (1981-2002) were directly assigned to the series of cumulative probabilities by MODIS (2002-2021). To restore the inputted values to their original magnitudes (NDVI), quantiles corresponding to their cumulative probabilities were computed using the MODIS NDVI. Therefore, the first 505 temporal layers of the harmonized AVHRR NDVI data, covering the period from 1981-07-01 to 2002-07-01, were merged with the MODIS

data spanning 2002-07-15 to 2012-01-01. The procedure was the same by integrating the VIIRS data, starting from 2012-01-15 and continuing up to the latest available data. This method ensures the dataset remains current, as it will be updated regularly with new VIIRS NDVI data releases.

### 3.2.2 Ratio-Based Harmonization

This harmonization approach involves calculating ratios between the NDVI values of two datasets during their overlapping

periods. Initially, we extracted the overlapping NDVI data from AVHRR and MODIS. For each corresponding semi-monthly interval within this overlap, we computed the average NDVI value from each dataset. The ratio of these average values—MODIS average divided by AVHRR average—provided a scaling factor that indicates how much the AVHRR data needed to be adjusted to match the MODIS data. Once the ratios were determined, we applied these scaling factors to the entire AVHRR dataset, adjusting each NDVI value to bring it into closer alignment with the corresponding MODIS values. This same

procedure was then applied to adjust the harmonized AVHRR-MODIS dataset to align with the VIIRS data.

### 3.3 kNDVI calculation

It has been well documented that a main limitation of the NDVI index as a function of vegetation activity stems from its inability to differentiate high biomass or dense vegetation areas accurately (Gao et al., 2023; Huang et al., 2021; Jinru and Su, 2017). To address this limitation, several studies have addressed the saturation effect by proposing adjustments to NDVI values

based on their relationship with specific vegetation properties, such as the Vegetation Fraction (VF) (Jiang and Huete, 2010; Liu et al., 2012) or by incorporating alternative vegetation indices to adjust NDVI values (Gu et al., 2013). Alternatively, to cope with the saturation issue, several indices have been proposed and compared with the NDVI performance (e.g., Aklilu Tesfaye and Gessesse Awoke, 2021). Among these alternatives, the Enhanced Vegetation Index (EVI) was developed to optimize the vegetation signal with improved sensitivity in high biomass regions (Huete et al., 2002) but it does not completely

solve the saturation problem (Camps-Valls et al., 2021). To this end, Camps-Valls et at. (2021) developed the kNDVI index, which transcends NDVI's limitations by leveraging higher-order spectral bands relationships. Thus, the kNDVI provides a higher accuracy in estimating vegetation parameters such as Leaf Area Index (LAI), Gross Primary Productivity (GPP), and chlorophyll content, outperforming NDVI in all biomes and climatic zones (Camps-Valls et al., 2021).

Considering the significant advantages of kNDVI, particularly its enhanced sensitivity for high-biomass estimation compared

to NDVI, we decided to include both kNDVI and its anomalies alongside NDVI data in our database.





To compute the kNDVI index we applied the simplified Eq. (1) as provided in Camps-Valls et al. (2021), allowing its computation directly from the harmonized NDVI time-series.

$$kNDVI = \tanh(\text{NDVI}^2), \tag{1}$$

where tanh represents the hyperbolic tangent function applied to the square of the NDVI values.

### 3.4 SNDVI and SkNDVI calculation

Standardizing data based on central tendency and dispersion statistics derived from NDVI time-series does not inherently yield measures that are comparable across different locations (Moesinger et al., 2022; Tam et al., 2023). While this method facilitates the identification of anomalies—deviations from the long-term average—within a specific location, it does not standardize these anomalies across varied ecosystems or geographical areas. Consequently, an anomaly of a certain magnitude in one location cannot be directly equated to an anomaly of the same magnitude elsewhere without accounting for the normal conditions specific to each area. In contrast, standardized indices transform these raw anomalies into a uniform scale, typically centered around zero and conforming to a Gaussian distribution. This approach is widely used in the calculation of drought indices (Laimighofer and Laaha, 2022; McKee et al., 1993; Stagge et al., 2015) and allows for direct comparisons of anomaly severity across diverse locations. To standardize the NDVI values, we followed the approach used for the reconstruction described in Section 3.2.1, where ten standardized series corresponding to ten different probability distributions were utilized for the long time series spanning from 1981 to 2023, using the distribution parameters over the whole period. The selection of the distribution for the series of each semi-monthly period, vegetation index, and pixel was based on the highest p-value obtained from the Shapiro-Wilks test among the ten distributions (Vicente-Serrano et al., 2024).

### 3.5 Time-series harmonization assessment

To evaluate the harmonization process of building the NDVI time-series, we used the Willmott's index of agreement (d) (Willmott, 1981, 1984; Willmott et al., 1985). The calculation of the d index was performed using the hydroGOF package in R (Zambrano-Bigiarini, 2024). In our study, the d index is utilized to assess the precision of the harmonization between AVHRR and MODIS NDVI data, and subsequently between the harmonized AVHRR-MODIS dataset and the VIIRS NDVI data. In practical terms, the index measures the cumulative differences between harmonized and reference NDVI values, against a hypothetical perfect model's predictions. Willmott's d index varies between 0 and 1, where a value of 0 indicates no agreement at all, and a value of 1 a perfect alignment between estimates and observations. The d index was applied on a per-pixel basis across AVHRR-MODIS and the integrated AVHRR-MODIS-VIIRS NDVI datasets, comparing them over their entire overlapping period with the reference data to determine an overall accuracy measure. Additionally, to examine temporal consistency, d index calculations were conducted for each semi-monthly interval, identifying potential seasonal performance shifts. The harmonization's efficacy was further explored across various land cover types (i.e., irrigated crops, non-irrigated crops, orchards, olive groves, vineyards, mixed vineyards-olive groves, meadows, grasslands, shrublands, mixed shrublands-

pastures, coniferous forests, riparian forests, eucalyptus crops, deciduous forests, mixed coniferous-eucalyptus, mixed coniferous-deciduous forests, unproductive lands and mixed vineyards-orchards), leveraging the 'Mapa de Cultivos y Aprovechamientos de España' (https://sig.mapama.gob.es/Docs/PDFServicios/Mapadecultivos.pdf, last accessed July 2024),

to identify discrepancies among different vegetation and land use categories. Given the pixel-by-pixel accuracy analysis, we also present d index maps, offering a spatially detailed view of the harmonization's effectiveness both overall and across seasons within the study area.

To complement the evaluation of data harmonization, we calculated the Pearson's correlation coefficient (r) and the Root Mean Square Error (RMSE) as measures of goodness of fit; the Percent Bias (PBIAS) as a measure of bias; and the ratio of standard

deviations (rSD) as a measure of bias in the variability of the distribution. These statistics were computed between the AVHRR-MODIS-VIIRS harmonized NDVI data and the original VIIRS NDVI data.

## 3.6 Near-real time monitoring system

The deployment of a web-based vegetation monitoring system significantly enhances management and decision-making capabilities in agriculture and environmental conservation. It provides tools for visualizing and downloading data, ensuring

timely access to essential information on vegetation condition and anomalies. This accessibility enables stakeholders—including farmers, researchers, and policymakers—to make informed decisions quickly, especially during critical periods of vegetation activity change. The system's user-friendly interfaces support interactive visualizations and the analysis of long-term trends, helping to identify potential environmental threats such as droughts, wildfires, or floods, and to assess agricultural practices.

To maintain this functionality, it is essential that the vegetation monitoring system is updated frequently, enabling real-time data analysis that allows stakeholders to promptly respond to changes across various landscapes. The database automatically updates every eight days when new VIIRS NDVI data become available. First, the incoming NDVI data are spatially and temporally resampled to seamlessly integrate into our database. These data are then used to compute and update the kNDVI index. Following the updates for NDVI and kNDVI, our approach specifically streamlines the integration of updates for

SNDVI and SkNDVI. For these standardized indices, we utilize the pre-calculated probability parameters established from a fixed reference period spanning from 1981–2023. These parameters support the computation of the standardized vegetation indices and are applied as new data arrives, ensuring seamless integration of updates without the need to recalculate the entire historical time series. This method prevents any alterations to previously established index values and accelerates the update process.





## 4 Results and Discussion

### 4.1 Harmonization assessment

To illustrate the outcome of the temporal harmonization process utilizing the ratio-based harmonization approach, Fig. 2 shows the transformation of the AVHRR data to align with MODIS NDVI data, and the subsequent transformation of the harmonized AVHRR-MODIS data to align with VIIRS NDVI data at one specific location (x=217700, y=4417800). Panel (a) of both Fig.2 and Fig. 3, showcase the disparities inherent in NDVI data obtained from different sensors. Notably, NDVI values from MODIS are systematically higher compared to those from AVHRR and VIIRS. These findings underscore the crucial need for data harmonization across sensors to establish a coherent and reliable NDVI time series, essential for accurate long-term monitoring of vegetation dynamics. After harmonization, a notably strong alignment between the AVHRR-MODIS and AVHRR-MODIS-VIIRS datasets is noticeable, where the d index values were 0.96 and 0.98, respectively, indicating a highly successful harmonization. Additional pixels are plotted and presented in the supplementary material (Fig. S1-S2).





**Figure 2. The ratio-based harmonization was applied to AVHRR Sp_1km_NDVI and MODIS MYD13A2 original data (a), resulting in AVHRR-MODIS aligned NDVI data (b). This dataset was subsequently harmonized with VIIRS NDVI data (c, d). Dates between dashed lines represent the overlapping period between datasets. Pixel coordinates: x= 217700, y=4417800.**


An initial visual comparison of the datasets harmonized using the quantile normalization and ratio-based methods reveals substantial performance differences. While both methods align NDVI data accurately during overlapping periods, the quantile normalization method's application of cumulative probabilities from these periods to earlier, non-overlapping segments tends

to reduce variability and trend consistency in the time series. This issue is particularly noticeable in pixels exhibiting clear trends, where NDVI distributions from overlapping periods differ markedly from those in non-overlapping periods. Conversely, the ratio-based harmonization method does not suffer from this problem, successfully preserving the original data variability and trends as illustrated in Fig. 3. Consequently, we discarded the quantile normalization method as a means of harmonization but continue to utilize it to standardize the NDVI database that was harmonized using the ratio-based approach.

This strategic application ensures that we leverage the strengths of both methodologies effectively.

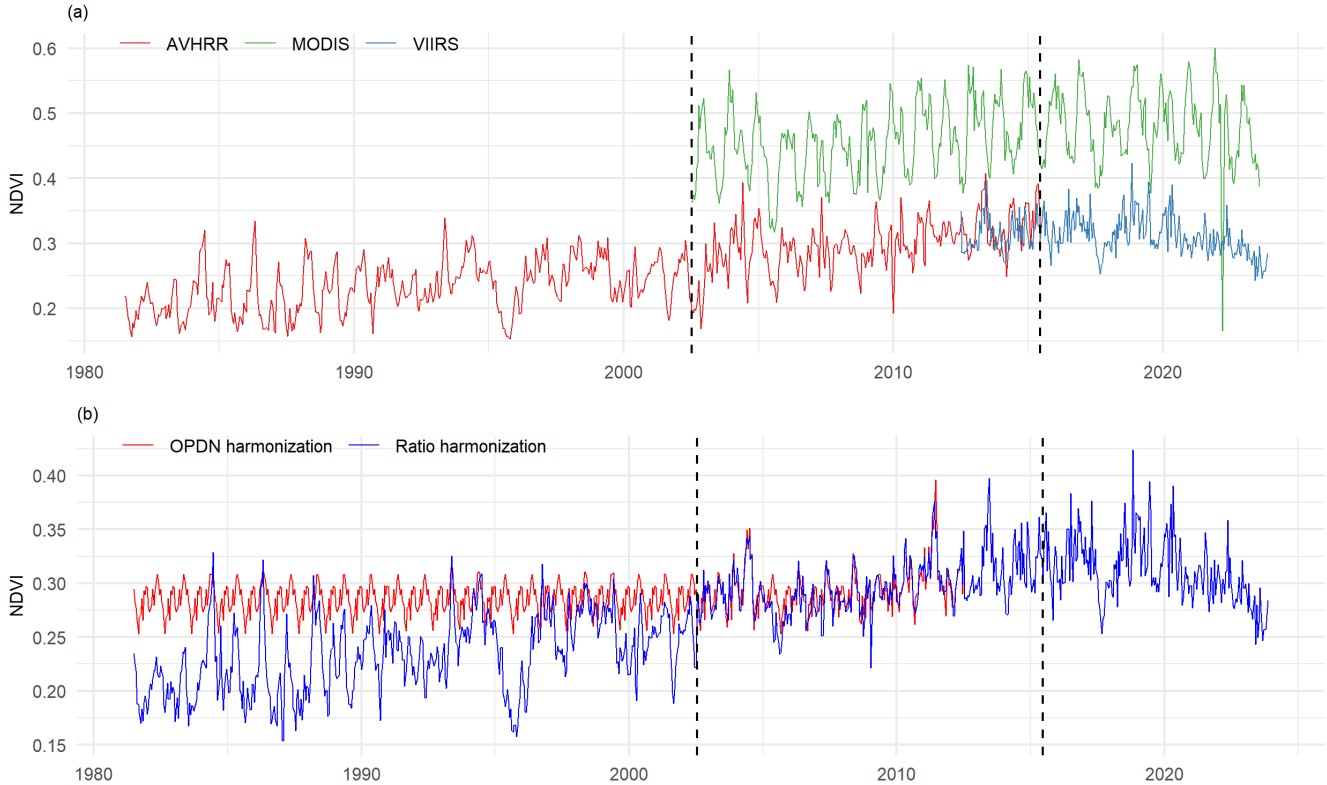

**Figure 3. Panel (a) shows the AVHRR Sp_1km_NDVI, MODIS MYD13A2 and VIIRS VNP13A2 NDVI data before harmonization. Panel (b) shows the harmonized NDVI data for both quantile normalization and ratio-based harmonization. For reference, dates between dashed lines represent the overlapping period between AVHRR and MODIS NDVI data. Pixel coordinates: x=438800, y=4146100.**



Earth System
Science
Data

The maps showing the spatial distribution of the d index values are depicted in Fig. 4. The panels (a), (b), and (c) evaluate the
harmonization between AVHRR and MODIS datasets, illustrating the overall annual agreement and the specific performance
for January and July months, respectively. The panels (d), (e), and (f) extend this evaluation to include the AVHRR-MODIS-
VIIRS datasets, following the same format for annual and seasonal assessment. Specifically focusing on the fully harmonized
AVHRR-MODIS-VIIRS NDVI time-series, a generally high annual agreement (Fig. 4d) is noted, characterized by d index
values predominantly above 0.7 across the Iberian Peninsula and Balearic Islands. Some regions, particularly along the Atlantic
coast of northern Spain, display comparatively lower agreement levels. The seasonal monthly maps suggest superior
harmonization performance in July (Fig. 4f) relative to January (Fig. 4e), revealing a consistent pattern in model performance
that corroborates observations from the annual agreement map. Notably, areas demonstrating lesser agreement are
predominantly located in the northern regions of Spain and Portugal.

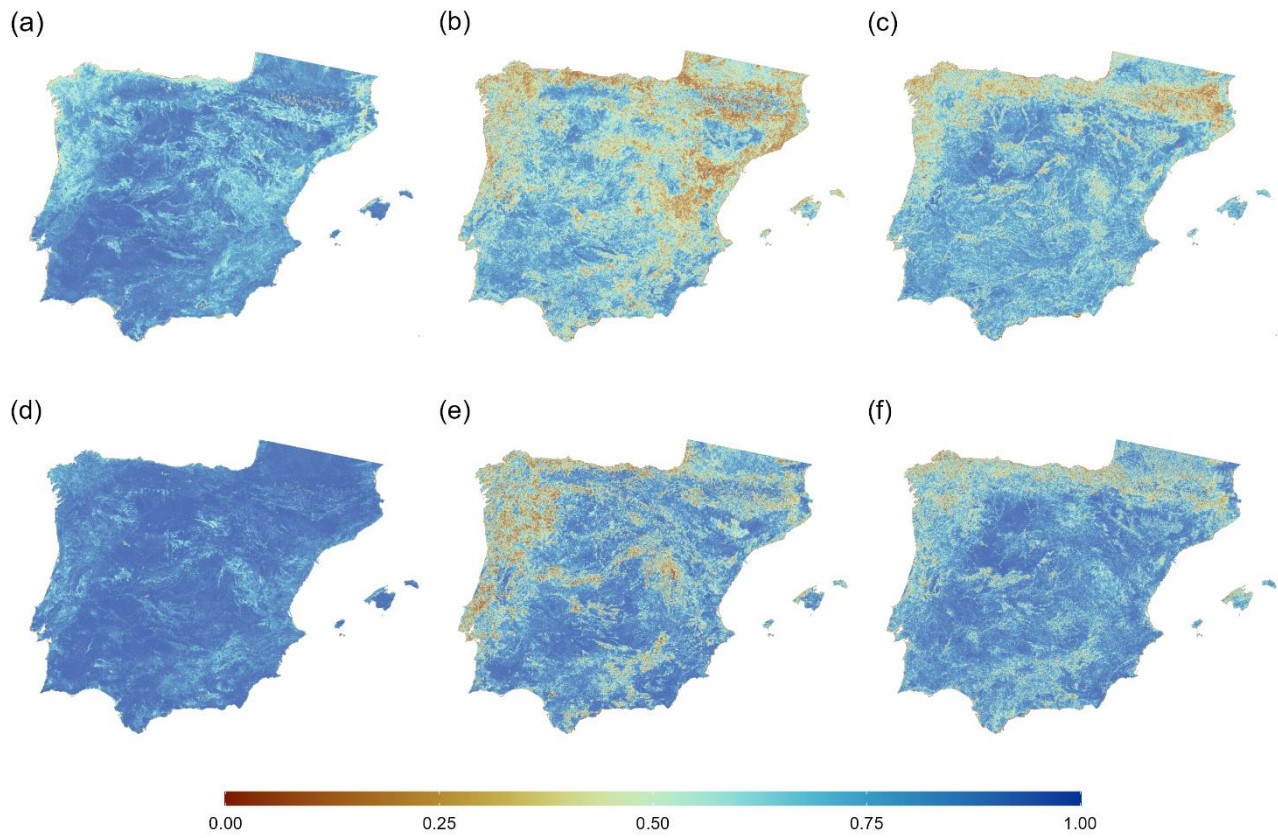


**Figure 4. Spatial distribution of Willmott's d index for NDVI time-series harmonization, calculated per pixel. Panel (a) displays the overall annual d index for the harmonized AVHRR-MODIS dataset. Panels (b) and (c) show the seasonal d index for the semi-monthly periods of December 01 and June 01, respectively. Panels (d), (e), and (f) correspond to the harmonized AVHRR-MODIS-VIIRS dataset, depicting the overall annual d index, and the seasonal d indices for December 01 and June 01 semi-monthly periods,**
**respectively.**



Figure 5a presents the Willmott's d index analysis across the entire period where harmonized AVHRR and MODIS NDVI data overlap. On average, the median d index for the entire overlapping period was 0.86. When examined by semi-monthly intervals, median d index values ranged between 0.57 and 0.75. These values tend to be slightly lower during autumn or winter months (September to January) and higher in summer, particularly in early and mid-June (median d=0.75). These seasonal

trends, indicating better model-data agreement during the summer months, may imply that the harmonization is more accurate during periods of active vegetation growth. The assessment of integrating harmonized AVHRR-MODIS NDVI data with VIIRS NDVI data is depicted in Fig. 5b. The temporal alignment performance (Fig. 5b) indicates a robust data reconciliation, with an overall median d index value of 0.95, and median d index values spanning from 0.74 and 0.83 for semi-monthly periods. Consistent with the patterns observed in the harmonization between AVHRR and MODIS NDVI data, the highest d

values are recorded in May and early June, with a median d index peaking at 0.83, while no substantial differences were found during the winter months.

The results of the assessment by land cover type between AVHRR-MODIS and VIIRS NDVI data are presented in the supplementary material (Fig. S3), revealing insights into seasonal and land cover-specific variations in harmonization performance. These differences highlight how NDVI data harmonization is influenced by land cover type and seasonality.

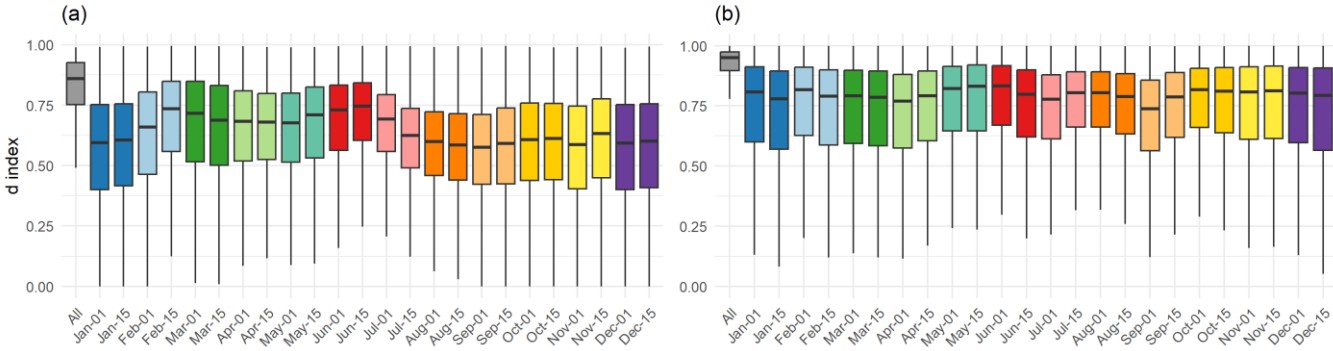


**Figure 5. Distribution of Willmott's d index values. Panel (a) shows the d index for harmonized AVHRR and MODIS NDVI data over the entire overlapping period, with median values ranging from 0.57 to 0.75, exhibiting seasonal variability. Panel (b) displays the d index for integrated AVHRR-MODIS and VIIRS NDVI data, with median values from 0.74 to 0.83, peaking in late spring and early summer.**

The statistics calculated to further assess the harmonization process, including Pearson's correlation coefficient, RMSE, PBIAS, and ratio of Standard Deviations (rSD), are presented in Fig. 6. The spatial distribution of Pearson's correlation coefficients indicates a predominance of strong positive correlations across the study area. This is evidenced by the extensive blue regions on the map and the high frequency of correlation values near 1 in the histogram. The RMSE values centered around 0.02, suggest that the harmonized NDVI data are highly accurate, with minimal deviation from the observed data. This

highlights the effectiveness of the harmonization process across the study area. The PBIAS values are very close to zero for all pixels. This is because the harmonization process employs a ratio to rescale one dataset to match the other, effectively eliminating systematic bias and resulting in no average tendency for the harmonized data to be consistently higher or lower


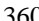

than the observed data. Finally, the results of the rSD statistic, with the majority of values near 1, suggest that the harmonization process was successful in preserving the variability of the original data.


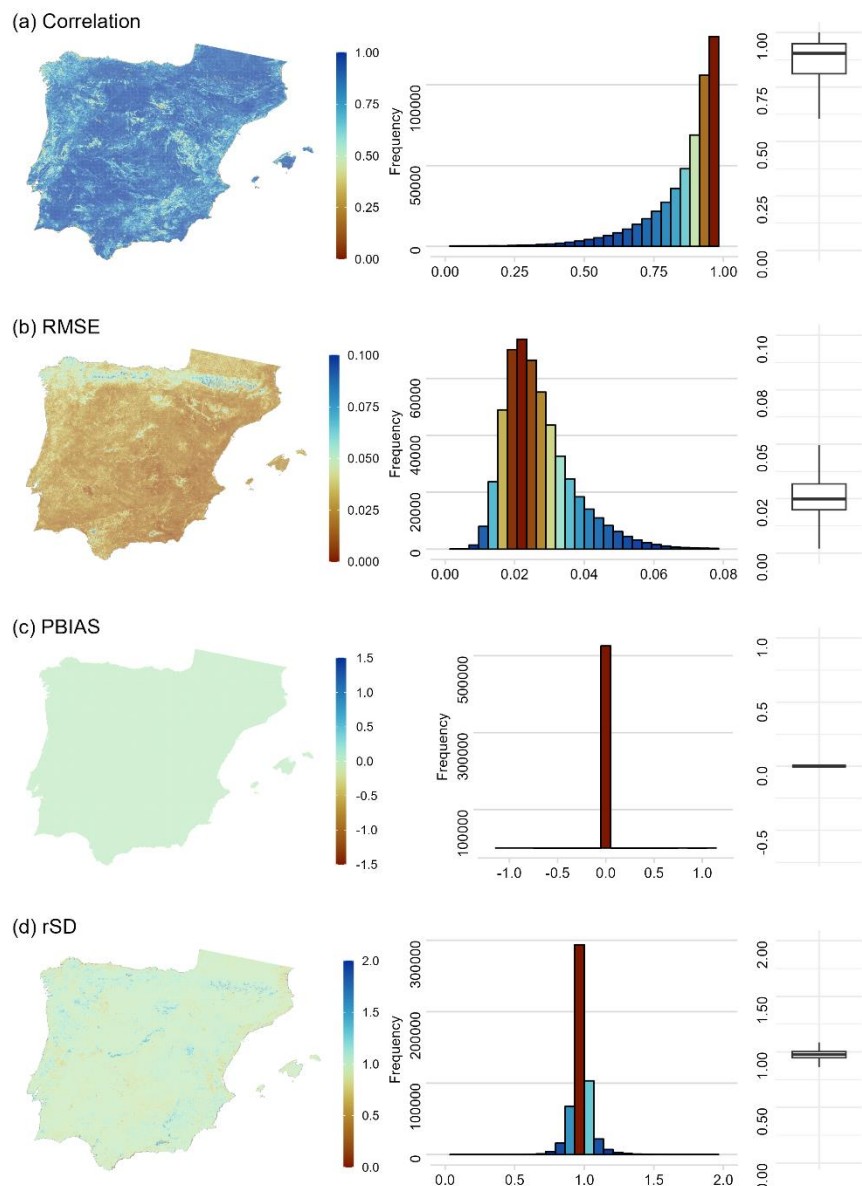

**Figure 6. Spatial and data distribution of the Pearson's Correlation (a), Root Mean Square Error (RMSE) (b), Percent Bias (PBIAS) (c) and ratio of Standard Deviations (rSD) (d), computed between the harmonized AVHRR-MODIS-VIIRS NDVI data and VIIRS NDVI data.**



## 4.2 NDVI-kNDVI time series

The maps in Fig. 7 illustrate the seasonal dynamics of vegetation cover as captured by the average of NDVI and kNDVI indices from 1981 to 2023. The spatial patterns of temporal correlation between the NDVI and kNDVI datasets, show very high correlation values across the Iberian Peninsula and the Balearic Islands for all seasons. However, during the winter season,

high mountainous areas exhibit reduced Pearson's r values, likely due to snow cover masking the vegetation signal. Conversely, in summer, when vegetative activity peaks in the northern peninsula, the indices are highly correlated, indicating consistent NDVI and kNDVI measurements.

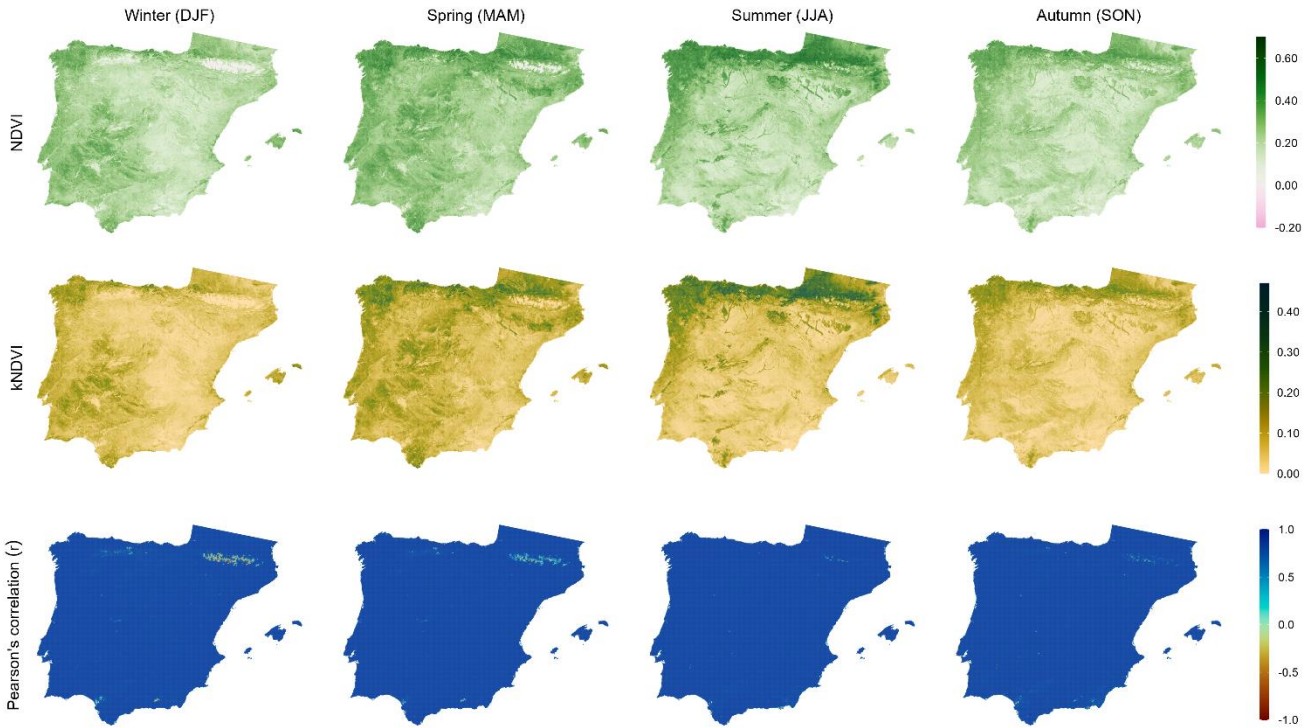

**Figure 7. Spatial distribution of seasonal NDVI and kNDVI average for the period 1981 to 2024 and Pearson's correlation**
**coefficient (r) between seasonal NDVI and kNDVI data.**

The scatter density plots in Fig. 8 provide a comparative analysis of the NDVI and kNDVI relationship across different seasons, as well as on an annual scale, from the period 1981 to 2023. The data reveal a linear relationship between NDVI and kNDVI, with R-squared values ranging from 0.87 in autumn to 0.95 in summer, indicating a strong correlation on both seasonal and annual scales. The observed regression slopes, approximately between 0.5 and 0.6, indicate that kNDVI typically rises at a

lesser rate than NDVI. It is also observed that kNDVI's rate of increase is more pronounced for NDVI values exceeding roughly 0.4-0.5, potentially due to NDVI's saturation at high biomass levels. Nonetheless, annual mean NDVI values surpassing 0.4-0.5 are rare, constituting less than 0.5% of the entire surface of the Iberian Peninsula and Balearic Islands (Fig. S4a). Summer exhibits the largest area (8%) with mean NDVI values above 0.5, likely as a result of high vegetation activity in northern Spain

during this season. Conversely, 46% of the region exhibited annual maximum NDVI values exceeding 0.5, primarily during

the spring and summer months when a substantial proportion of the land—38% in spring and 22% in summer—experienced

peak NDVI values above 0.5 (Fig S3b). Therefore, employing kNDVI could potentially be more advantageous for monitoring

vegetation during periods and areas where peak NDVI values indicate high vegetation activity.



**Figure 8. Relationship between NDVI and kNDVI at seasonal and annual scales. The sample size used for the density plots was n= 10,000.**

### 4.3 Standardized NDVI and kNDVI time-series

The use of standardized vegetation indices, such as SNDVI and SkNDVI, provides powerful tools for analysing and detecting

environmental disturbances over both short and long-term periods. Here we highlight some of these applications.

The temporal distribution of the SNDVI across the entire time-series (1981-2023) is illustrated in Fig. 9. It displays the

percentage of land surface for different SNDVI categories (from mild to extremely severe negative and positive anomalies)

for the Iberian Peninsula and the Balearic Island at semi-monthly scale. This comprehensive view allows for the identification of periods with significant vegetation stress or growth, providing insights into how different environmental factors have influenced vegetation dynamics over the decades. For instance, the early 1980s to mid-1980s show recurring severe negative

anomalies, indicating widespread adverse conditions. Another notable period of severe negative anomalies is observed in the late 1990s to early 2000s. The severe drought of 2005, which impacted a vast portion of the Iberian Peninsula (García-Herrera et al., 2007), is clearly reflected in the data (Fig. S5). Furthermore, the patterns of severe negative anomalies observed in the SNDVI time-series are consistent with drought episodes identified using climatic indices such as the Standardized Precipitation-Evapotranspiration Index (SPEI) and the Standardized Precipitation Index (SPI) (González-Hidalgo et al., 2018).

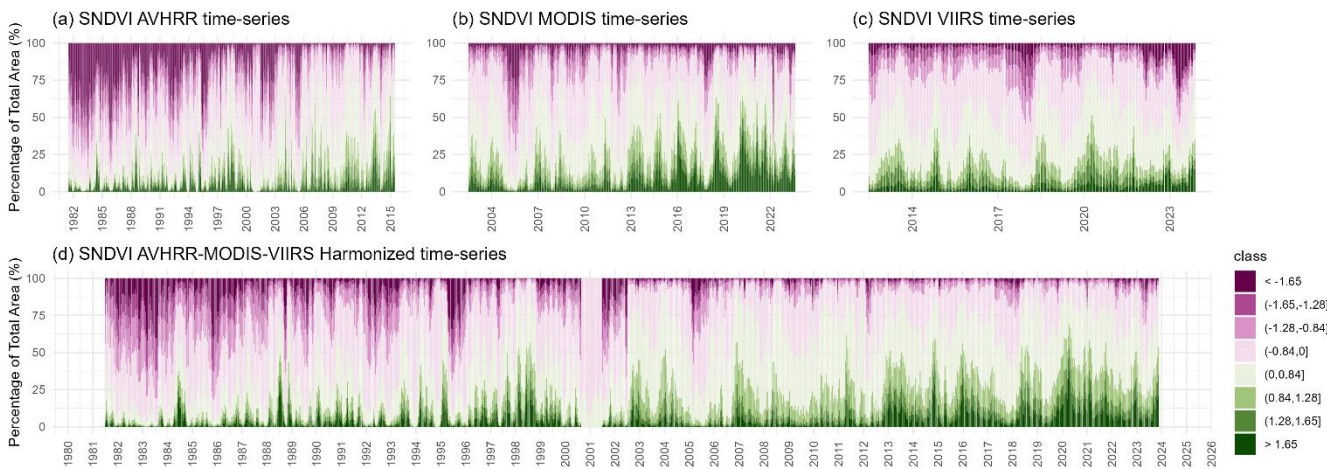


**Figure 9. Temporal patterns of Standardized NDVI (SNDVI) categories in the Iberian Peninsula and Balearic Islands from 1981 to 2023, as a percentage of total land area. Panels (a), (b), and (c) depict the SNDVI calculated independently for the AVHRR, MODIS, and VIIRS time-series respectively, while panel (d) shows the SNDVI calculated from the entire harmonized NDVI time-series. The color gradient from purple to dark green marks the range of SNDVI values, with purple representing the lowest (< -1.65) and dark**

**green the highest (> 1.65), capturing the historical vegetation dynamics within the region.**

Figure 10 presents the SNDVI time-series for different land cover types, including non-irrigated land, irrigated land, coniferous forests, broad-leaf forests, and pastures. Each subplot includes the location of the respective land cover type on the map and the corresponding SNDVI time-series. This figure provides a detailed view of how different land cover types respond to

environmental disturbances, highlighting the variability in vegetation response across different ecosystems.



**Figure 10. Standardized NDVI and kNDVI time series at different locations and land covers.**






As an illustrative example of NDVI anomalies, Fig. 11 showcases significant environmental events across the study period. Specifically, Fig. 11a highlights the NDVI anomalies during late 2017 (October 2017), a period marked by one of the most severe drought events to impact Europe in recent decades, with the Iberian Peninsula being particularly affected (García-

Herrera et al., 2019). The NDVI anomalies are expressed as standard deviations from the mean NDVI of the entire time series. The map reveals areas with very low standard deviation values (<-3), indicating significant deviations from typical NDVI values, which are indicative of substantial vegetation stress or decline during this extreme drought event. Additionally, exceptionally low anomaly values are detected in central-northern Portugal and Galicia, regions that suffered from catastrophic wildfire outbreaks within the same year (Llorens et al., 2021). Figure 11b illustrates the NDVI anomalies during the first half

of January 2021, highlighting the areas impacted by the storm Filomena. The most affected areas align with regions that experienced significant and historic snowfall in early January, particularly on the 8th and 9th, where snow accumulation reached unprecedented levels in various regions (https://www.aemet.es/es/conocermas/borrascas/2020-2021/estudios_e_impactos/filomena , last accessed July 2024). The Standardized NDVI's ability to capture these variations underscores its utility in monitoring the impact of severe climatic events on vegetation cover.

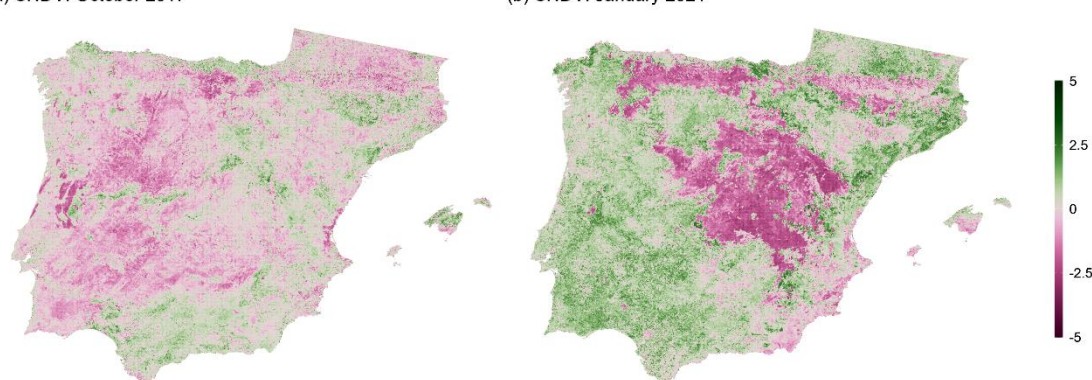


**Figure 11. SNDVI from significant climatic events. Panel (a) shows the impact of the severe drought and wildfires on the Iberian Peninsula in October 2017, with notable vegetation stress indicated by standard deviation values below -3. Panel (b) presents SNDVI from the storm Filomena in January 2021, highlighting areas with unprecedented snowfall.**

### 4.4 Vegetation Monitoring System

The Long-term Remote Sensing Vegetation Monitoring System is available at https://vi-anomalies.csic.es. The web tool enables users to select a vegetation index (NDVI, KNDVI, SNDVI and SkNDVI) for mapping and choose the date of interest. The home page displays the most recent data by default; however, users can select any date from 1981 to the present. This feature allows users to visualize and compare vegetation conditions at various points in the time series. Specifically, visualizing SNDVI or SkNDVI is particularly useful for direct comparison across different regions and dates. Users can zoom in on maps,

query values at specific cursor locations, and change the date using a time slider or a date picker. Additionally, the web tool allows the selection of a point on the map, and then, automatically, a plot is displayed with the time series of the selected vegetation index at the chosen grid cell. The plot is also interactive as the user may consult the index values at any point of the

time series and zoom to specific periods. Finally, there is a button to download the time series of the selected coordinates. The data is downloaded in text format as comma-separated values (csv), including the data series of the selected index. It is also

possible to download the complete NetCDF file of the vegetation index previously selected in the Index button selector.

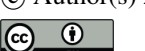

**Figure 12. Web-tool for the long-term vegetation monitoring system in the Iberian Peninsula and Balearic Islands. The bottom figures illustrate an example of NDVI and SNDVI temporal evolution at a specific location (40.18, -7.92).**

**5 Data availability**

The long-term NDVI and kNDVI database, along with their standardized versions, SNDVI and SkNDVI, are publicly available at the DIGITAL.CSIC repository via the following link: https://doi.org/10.20350/digitalCSIC/16201 (Franquesa et al., 2024). Access to regular database updates is available at https://vi-anomalies.csic.es, where data can be visualized and downloaded.

The database covers the entire Iberian Peninsula (including Spain, Portugal, and the French Pyrenees) and the Balearic Islands,
providing data at a semi-monthly temporal resolution and 1.1 km spatial resolution from 1981 to the present. To maintain the
coordinate reference system (crs) of the original AVHRR NDVI dataset, the data is provided in the ED50 UTM30N
(EPSG:23030) projected crs, in NetCDF format.

## 6 Conclusions

This study presents a database of vegetation indices (NDVI and kNDVI), along with their standardized versions (SNDVI and
SkNDVI), tailored for the detection and analysis of vegetation activity anomalies across the entire Iberian Peninsula and
Balearic Islands. By integrating data from three distinct sensors—AVHRR, MODIS, and VIIRS—this database provides an
unprecedented long-term series that spans more than 40 years, from 1981 to the present. It stands out for its regular near-real-
time updates (semi-monthly), making it an indispensable tool for vegetation monitoring. Significantly, the SNDVI and
SkNDVI datasets represent the only database of its kind dedicated to vegetation activity anomalies for the Iberian Peninsula
and Balearic Islands, underlining the critical importance of such data for environmental monitoring, including risk assessment,
and showcasing its crucial applications in agriculture. The harmonization quality between AVHRR-MODIS and VIIRS NDVI
data, rigorously assessed using the Willmott's index of agreement (d), which indicated high overall temporal and spatial
performance (d=0.95), further emphasizes the reliability and utility of this unique resource. The exceptional value of the
SNDVI and SkNDVI database in supporting environmental and agricultural decision-making processes cannot be overstated,
offering unparalleled insights for managing vegetation health and productivity.

### Author contributions

MF led the production of the paper and contributed to the conceptualization of the study, generated and evaluated the
harmonized database and wrote the initial draft. FR produced the standardized vegetation indices, developed the web-monitor
system, and automated the data updates. MA and DV developed the web-monitor system. MAM, AHM and SB contributed to
the writing and review. SMVS, serving as the project's scientific leader, created the harmonization and standardization
algorithms, coordinated the overall execution of the project, and recommended the preparation of this article. All authors
contributed to the writing and review of the manuscript and approved its final version.

### Competing interests

The authors declare that they have no conflict of interest.

### Acknowledgements

To enhance the clarity and quality of the English language in this article, AI tools were utilized.





**Financial support**

This research has been supported by the Grant JDC2022-048710-I funded by MCIN/AEI/ 10.13039/501100011033 and by the European Union NextGenerationEU/PRTR.

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
