# Peer review of "Near-real-time vegetation monitoring and historical database (1981present) for the Iberian Peninsula and the Balearic Islands"

_Earth System Science Data, 2024_

## Referee Comment (RC2)

The manuscript ESSD-2024-351, presents a near real-time vegetation monitoring system for Spain, spanning a historical database from 1981 to the present. It is a well-written and comprehensive study that clearly explains the data sources, preprocessing methods, results, and validation procedures. The integration of harmonized satellite imagery from AVHRR, MODIS, and VIIRS using widely accepted vegetation indices is a commendable and novel approach. This work holds significant value for understanding vegetation dynamics in the context of climate change.

However, before publication, the following technical corrections and clarifications should be addressed:

➤ **Incomplete Documentation**:
The README file provided at [https://digital.csic.es/handle/10261/353068](https://digital.csic.es/handle/10261/353068) is incomplete and confusing. Please revise and complete the document to ensure clarity and usability for future users.

➤ **Broken URLs**:
Several URLs in the manuscript are not functioning correctly:

- Page 4: The link https://e4ftl01.cr.usgs.gov/MOLA/MYD13A2.061 is broken. Please remove or replace it with a working alternative.
- Page 5: The link https://lpdaac.usgs.gov/products/vnp13a2v002/ is also not working. Kindly verify and update all URLs to avoid confusion.

➤ **Clarification on Limitations**:
The manuscript reviews previous vegetation indices methods but does not elaborate on the limitations of the generated database, except for the temporal limitation. A brief discussion on why these datasets may not be suitable for the Iberian Peninsula and Balearic Islands would help clarify the motivation behind the proposed method and its near-real-time capabilities.

➤ **Figure Reference Error**:
On page 17, the manuscript refers to Figure S3b, which does not exist in the supplementary materials. It appears the intended reference is to Figure S4. Please correct this.

➤ **Clarifications on Figures and Results**:

- **Figure 3**: Please explain why OPDN harmonization does not overlap with Ratio harmonization before the year 2000. Also, define the acronym "OPDN" clearly.
- **Figure 4**: The annual d index appears notably lower in the northern regions of Spain and Portugal. Please provide a clearer explanation and explicitly define each subfigure.
- **Figure S3**: There is a noticeable difference in the seasonal median values for grasslands and Eucalyptus compared to dry crops. Could phenological characteristics be influencing these variations? This is important for assessing the robustness of the method for global applications.

- **Figure 7**: The manuscript states that Pearson's correlation values are low in winter and higher in summer. However, low values are also observed in May. This could be due to spring snowmelt and early vegetation bloom, which should be mentioned for clarity.
- **Figure 9**: The vegetation anomalies corresponding to the 2005 drought are well captured. However, please elaborate on the causes of prolonged drought conditions during the 1980s and again in 2002.
- **Figure 10**: The observed vegetation variability is a key highlight. Please provide ground truth evidence or reasoning to support these findings. Has the use of GEDI LiDAR data been considered for validating vegetation responses?

Overall, this manuscript presents a valuable and innovative contribution to the field of vegetation monitoring and climate research. The integration of harmonized satellite data and vegetation indices across multiple decades for Spain is both timely and impactful. With minor technical corrections and clarifications as outlined above, the manuscript will be well-positioned for publication and will serve as a useful resource for researchers and practitioners working on environmental monitoring and land surface dynamics.

---

## Author Comment (AC1)

**Response to Reviewer #1**

The authors wish to thank anonymous referee #1 for their kind words and positive evaluation of our work. We are pleased to know that both the manuscript and the dataset have been considered relevant, timely, and useful for a wide range of potential users. We have carefully considered the comments and revised the manuscript accordingly.

Please, find a detailed response to the comments below. Original comments are in black and our response is in blue.

**\*\**

Near-real-time vegetation monitoring and historical database (1981-present) for the Iberian Peninsula and the Balearic Islands

This work and its associated dataset include spatio-temporal series of vegetation indices for Iberian Peninsula and Baleraric Islands during 1981 to date. The authors also provide a link to visualize and download the trend of the indices. All of these resources are highly relevant for researchers, managers, land-owners, policy-makers as well as other stakeholders working on forestry, agriculture and other environmental sciences. Therefore, both the paper and the data are highly timely and potentially useful.

Overall, the manuscript is well-written and it is easy to follow. I recommend accepting it for publication.

Below, you can find some minor comments:

L107-108: why do the layer represent the highest value? Why not the mean or the median?

We thank the reviewer for this pertinent question. The layers represent the maximum NDVI value within each compositing period because we applied the Maximum Value Composite (MVC) method, which is a widely adopted technique in remote sensing for vegetation monitoring (Holben, 1986). This approach minimizes the effects of cloud contamination, atmospheric interference, and low solar angle conditions, which typically lead to underestimated NDVI values. By selecting the maximum value, we ensure that the NDVI is more representative of actual vegetation activity during the period, as it is less affected by transient atmospheric or observational artifacts. In contrast, using the mean or median could retain such distortions, leading to less reliable indicators of vegetation health.

L112: please add the EPSG as you do at the end of the manuscript.

We have now added the EPSG code (EPSG:23030) to Line 112 to ensure consistency with the information provided at the end of the manuscript.

L144: what is the difference between the average here and the highest NDVI in L107-108.

We appreciate the reviewer's question and the opportunity to clarify this point. The difference lies in the compositing method used and the stage of data processing. In Lines 107–108, we refer to the Maximum Value Composite (MVC) method applied to the AVHRR dataset, where the highest NDVI value within each semi-monthly period is selected to minimize cloud and atmospheric effects.

In contrast, at Line 144, we are referring to the linear interpolation process used to temporally resample MODIS and VIIRS data. In this case, daily NDVI values are estimated through interpolation, and the average NDVI is then calculated over the same semi-monthly periods used in the AVHRR data. This ensures temporal alignment across sensors. Therefore, while the AVHRR

uses the maximum observed value in each period, the MODIS and VIIRS NDVI values represent the average of interpolated daily values.

This difference reflects the original compositing methods and data availability for each sensor, and our harmonization approach accounts for these differences to ensure consistency in the final time series.

L254: I have some concerns on the map used. In my opinion, authors should provide some details such as map spatial resolution. On the other hand, is the map temporally dynamic? I mean, do the land uses change over time? I wonder how changes in land uses (or the lack of the changes in the map) have been considered in your study.

Thank you for this insightful observation. The land cover information used in our assessment is based on the Mapa de Cultivos y Aprovechamientos de España (MCA), developed by the Spanish Ministry of Agriculture. The version used in our study corresponds to the 1980–1990 edition, which was digitized in the 1990s and has a cartographic scale of 1:50,000. The minimum mapping unit is a polygon (patch), representing areas of homogeneous land cover. We have now added this information to the manuscript to clarify the spatial resolution of the dataset.

Regarding temporal dynamics, the MCA is not a time-varying product and therefore reflects land use for the reference period of the 1980s. Consequently, it does not capture land cover changes that may have occurred during the full study period (1981–present). However, the goal of this analysis was to assess the performance of the NDVI harmonization across different land cover types, not to quantify land use changes. We acknowledge that future work could benefit from integrating dynamic land cover datasets to account for land use transitions over time.

We have modified the paragraph in the manuscript as follows (new text in bold italics):

The harmonization's efficacy was further explored across various land cover types (i.e., irrigated crops, non-irrigated crops, orchards, olive groves, vineyards, mixed vineyards-olive groves, meadows, grasslands, shrublands, mixed shrublands-pastures, coniferous forests, riparian forests, eucalyptus crops, deciduous forests, mixed coniferous-eucalyptus, mixed coniferousdeciduous forests, unproductive lands and mixed vineyards-orchards), leveraging the Mapa de **Aprovechamientos** Cultivos de y España (https://sig.mapama.gob.es/Docs/PDFServicios/Mapadecultivos.pdf, last accessed July 2024), to identify discrepancies among different vegetation and land use categories. This land cover dataset corresponds to the 1980–1990 period and was digitized in the 1990s, with a cartographic scale of 1:50,000. Each mapping unit, or patch, represents a polygon with homogeneous land use. As the MCA is not temporally dynamic, changes in land use over the study period were not accounted for. Nevertheless, it provides a consistent reference for evaluating harmonization performance across major land cover classes. Given the pixel-bypixel accuracy analysis, we also present d index maps, offering a spatially detailed view of the harmonization's effectiveness both overall and across seasons within the study area.

L365. The color ramps of the maps do not correspond with the color ramps of the histogram. I mean, for instance in Fig 6a, correlation values equal to 1 are represented in blue in the map but these values are red in the histogram. Please, be consistent.

Thank you for raising this issue. We have revised all histograms to ensure full visual consistency with the corresponding maps. Specifically, we now apply the same continuous color palette (roma) to both maps and histograms. Each histogram bar is colored according to the value it represents, using the same color scale as in the maps. This ensures that high or low values are

visually represented in the same way across both spatial and statistical plots, thus improving interpretability and consistency.

Please find the revised figure below:

---

## Author Comment (AC2)

Response to Reviewer #2

The authors wish to thank anonymous referee #2 for their positive evaluation of our work. We have carefully considered the comments and revised the manuscript accordingly.

Please, find a detailed response to the comments below. Original comments are in black and our response is in blue.

**

*The manuscript ESSD-2024-351, presents a near real-time vegetation monitoring system for Spain, spanning a historical database from 1981 to the present. It is a well-written and comprehensive study that clearly explains the data sources, preprocessing methods, results, and validation procedures. The integration of harmonized satellite imagery from AVHRR, MODIS, and VIIRS using widely accepted vegetation indices is a commendable and novel approach. This work holds significant value for understanding vegetation dynamics in the context of climate change.*

*However, before publication, the following technical corrections and clarifications should be addressed:*

**Incomplete Documentation:**

The README file provided at https://digital.csic.es/handle/10261/353068 is incomplete and confusing. Please revise and complete the document to ensure clarity and usability for future users.

We acknowledge the reviewer's observation that the README file could be improved for clarity. The file currently follows the standard template provided by the DIGITAL.CSIC curators and is intended to supply the essential contextual information on the dataset (authors, temporal and spatial coverage, file structure, citation, etc.). More detailed methodological descriptions and procedures are provided in the associated publication, which is directly linked from the repository entry.

**Broken URLs:**

Several URLs in the manuscript are not functioning correctly:

• Page 4: The link https://e4ftl01.cr.usgs.gov/MOLA/MYD13A2.061 is broken.

Please remove or replace it with a working alternative.

• Page 5: The link https://lpdaac.usgs.gov/products/vnp13a2v002/ is also not

working. Kindly verify and update all URLs to avoid confusion.

Thank you for your observation. We have updated the broken URLs and verified that all links in the manuscript are currently functional.

**Clarification on Limitations:**

The manuscript reviews previous vegetation indices methods but does not elaborate on the limitations of the generated database, except for the temporal limitation. A brief discussion on why these datasets may not be suitable for the Iberian Peninsula and Balearic Islands would help clarify the motivation behind the proposed method and its near-real-time capabilities.

We thank the reviewer for this helpful suggestion. In fact, the limitations of existing NDVI products are already discussed in the Introduction. We note, for example, that the coarse spatial resolution of GIMMS (over 9 km), the limited temporal coverage of MODIS (since 2000) and VIIRS (since 2012), and the restricted availability of regional AVHRR-based products reduce the

suitability of these datasets for heterogeneous landscapes such as the Iberian Peninsula and the Balearic Islands. We also highlight that the Sp_1km_NDVI dataset, while more appropriate for the region, only extends to 2015:

*"The coarse spatial resolution (over 9 km for GIMMS NDVI datasets) or the limited temporal coverage of MODIS (since 2000) and VIIRS (since 2012) NDVI datasets restrict their applicability, most notably in regions with highly fragmented and diverse landscapes. … For the Iberian Peninsula and the Balearic Islands, the AVHRR NDVI product, Sp_1km_NDVI, (1981 to 2015, 1.1 km resolution) is available but the time-series only extends until 2015 (Vicente-Serrano et al., 2020)."*

These aspects introduce the main limitations of existing NDVI products for the region and clearly illustrate the need for a harmonized, near-real-time database. Our generated database addresses these constraints by providing long-term temporal continuity, medium spatial resolution, and operational updates, thereby overcoming the most relevant limitations identified above.

**Figure Reference Error:**

On page 17, the manuscript refers to Figure S3b, which does not exist in the supplementary materials. It appears the intended reference is to Figure S4. Please correct this.

Thank you, we have corrected this error.

**Clarifications on Figures and Results:**

Figure 3: Please explain why OPDN harmonization does not overlap with Ratio harmonization before the year 2000. Also, define the acronym "OPDN" clearly.

We thank the reviewer for pointing this out. The divergence between Quantile Normalization and Ratio harmonization before 2000 arises from methodological differences. Quantile Normalization relies on the statistical distributions of the overlapping periods (2002–2015, 2012–2021) to normalize earlier data. Because NDVI values from the 1980s and 1990s differ from those in the overlap years, this approach tends to reduce variability and distort long-term trends, producing inconsistencies before 2000. In contrast, the ratio-based method applies a scaling factor that better preserves the original variability and trends across the entire record. This explains why both methods agree during the overlap but diverge in the earlier part of the time series.

Since this explanation is already detailed in the manuscript, we have added the following clarifying sentence to the caption of Fig. 3:

*"Divergence before 2000 reflects the limitations of quantile normalization, which relies on distributions from the overlapping period (2002–2015, 2012–2021) and therefore reduces variability and distorts long-term trends, unlike the ratio-based method."*

Additionally, we removed the acronym OPDN from the figure, as it is not used in the manuscript.

Figure 4: The annual d index appears notably lower in the northern regions of Spain and Portugal. Please provide a clearer explanation and explicitly define each subfigure.

We thank the reviewer for this observation. In Fig. 4, panels (a–c) show the Willmott's d index for the AVHRR–MODIS harmonization (annual, December 1, and June 1 semi-monthly periods, respectively), while panels (d–f) show the same for the full AVHRR–MODIS–VIIRS harmonization. The lower d values observed in the northern regions of Spain and Portugal, as well as in

mountain areas are likely related to persistent cloud cover, frequent snow, and the prevalence of dormant vegetation during winter. This reduced activity means that differences across sensors are less relevant in ecological terms, since the most critical periods for monitoring vegetation dynamics are those of maximum activity, when harmonization performance is notably higher.

To illustrate this, we include two screenshots from our web viewer showing NDVI in December (low activity) and June (high activity).

*December*

[Figure]

*June*

[Figure]

To clarify this point in the manuscript, we have explicitly defined each subfigure in the caption and included a brief explanation of why lower d values in northern and mountainous regions are of limited practical significance.

Figure S3: There is a noticeable difference in the seasonal median values for grasslands and Eucalyptus compared to dry crops. Could phenological characteristics be influencing these variations? This is important for assessing the robustness of the method for global applications.

This is indeed a very interesting comment. In this work, we did not specifically analyze the causes associated with different land cover types that may influence the harmonization process. We agree that phenological characteristics are likely to play a role, as also noted in our previous response regarding vegetation dormancy, and this would be an interesting subject for further research. In addition to vegetation phenology, the geographic distribution of these land cover types is also likely to influence the intra-annual variability observed in the d index distributions. For example, eucalyptus plantations are mainly located in northern Spain, where cloud cover is more frequent, which may contribute to the observed patterns. In any case, this topic would require a more detailed and specific study to be properly addressed.

We have slightly modified the manuscript as follows:

"*The results of the assessment by land cover type between AVHRR-MODIS and VIIRS NDVI data are presented in the supplementary material (Fig. S3), revealing insights into seasonal and land cover-specific variations in harmonization performance. These differences highlight how NDVI data harmonization is influenced by land cover type and seasonality**, and may be partly***

*explained by phenological characteristics and the geographic distribution of the land cover types, although a more detailed analysis would be required to fully disentangle these effects.*"

Figure 7: The manuscript states that Pearson's correlation values are low in winter and higher in summer. However, low values are also observed in May. This could be due to spring snowmelt and early vegetation bloom, which should be mentioned for clarity.

As shown in Fig. 7 for the March–April–May (MAM) period, lower correlation values are also observed in high mountain regions such as the Pyrenees, Picos de Europa, and Sierra Nevada in southern Spain. These spatial patterns coincide with those found in winter and are similarly related to snow accumulation, which may persist well into spring at higher elevations. This effect is illustrated in the additional figures from our web viewer (red areas are snow covered).

[Figure]

Accordingly, we have modified the paragraph of the manuscript as follows:

*"The maps in Fig. 7 illustrate the seasonal dynamics of vegetation cover as captured by the average of NDVI and kNDVI indices from 1981 to 2023. The spatial patterns of temporal correlation between the NDVI and kNDVI datasets, show very high correlation values across the Iberian Peninsula and the Balearic Islands for all seasons. However, during the winter season **and early spring**, high mountainous areas exhibit reduced Pearson's r values, likely due to snow cover that can mask the vegetation signal. Conversely, in summer, when vegetative activity peaks in the northern peninsula, the indices are highly correlated, indicating consistent NDVI and kNDVI measurements."*

Figure 9: The vegetation anomalies corresponding to the 2005 drought are well captured. However, please elaborate on the causes of prolonged drought conditions during the 1980s and again in 2002.

Thank you for this comment. We have added more details in the discussion, specifying the causes of the drought periods and complementing them with new references. The paragraph has been modified as follows:

*"For instance, **several drought-related anomalies are evident in the early to mid-1980s, followed by a sequence of negative anomalies between the late 1980s and mid-1990s. Additional widespread anomalies are observed in the early 2000s, particularly during 2001–2002. These periods correspond to phases characterized by negative precipitation anomalies over the Iberian Peninsula, largely driven by a deficit of rainfall during the cold season caused by the strengthening of the North Atlantic Oscillation (NAO), which has been extensively***

*documented in the scientific literature (Trigo et al., 2004). The severe drought of 2005, which impacted a vast portion of the Iberian Peninsula (García-Herrera et al., 2007), is clearly reflected in the data (Fig. S5). Furthermore, the timing and magnitude of severe negative anomalies observed in the SNDVI time-series are consistent with drought episodes identified using climatic indices such as the Standardized Precipitation-Evapotranspiration Index (SPEI) and the Standardized Precipitation Index (SPI) (González-Hidalgo et al., 2018)* **and with the periods identified in the catalogue of drought events in peninsular Spain (Trullenque-Blanco et al., 2024)***."*

Figure 10: The observed vegetation variability is a key highlight. Please provide ground truth evidence or reasoning to support these findings. Has the use of GEDI LiDAR data been considered for validating vegetation responses?

We understand that validation is crucial to provide users with information about product accuracy. In our study, we applied harmonization and standardization procedures and evaluated their performance through statistical methods. The NDVI products used to build the long-term dataset have already been extensively validated, as described in the Data section.

We did not consider GEDI LiDAR as a validation source because GEDI data are only available from 2019 onward and therefore do not overlap with most of our long-term NDVI record, which begins in the 1980s.

Regarding accuracy, the original manuscript already included the following information, which we believe provides the necessary details about the current validation status of the three datasets used:

Sp_1km_NDVI

*"The accuracy of the Sp_1km_NDVI dataset was assessed by seasonal and annual comparison with other three widely used global NDVI products: GIMMS3g, SMN, and MODIS. These comparisons revealed similar spatial and temporal patterns to those observed in the referenced products (Vicente-Serrano et al., 2020)."*

MYD13A2 C61

*"The dataset has been validated to Stage 3 according to the validation stages defined by the Land Product Validation (LPV) subgroup (https://lpvs.gsfc.nasa.gov/, last access in September, 2025), ensuring high reliability with an accuracy of ±0.025 for under ideal conditions. These error estimates are based on rigorous comparisons with ground-based and other satellite data across diverse ecosystems (http://tinyurl.com/zt3uttab, last access in September, 2025). "*

VNP13A2 V002

*"VIIRS-based NASA VIs have achieved Stage 1 validation according to the LPV subgroup standards. Preliminary assessments of VIIRS VIs through comparison with ground-based data collected at AErosol RObotic NETwork (AERONET) sites indicated an overall accuracy of 0.009 for the Top Of Canopy (TOC) NDVI (Shabanov et al., 2015). This accuracy represents the average deviation of the measured values from the true or reference values."*

Overall, this manuscript presents a valuable and innovative contribution to the field of vegetation monitoring and climate research. The integration of harmonized satellite data and vegetation indices across multiple decades for Spain is both timely and impactful. With minor technical corrections and clarifications as outlined above, the manuscript will be well-positioned

for publication and will serve as a useful resource for researchers and practitioners working on environmental monitoring and land surface dynamics.

---

## Author Response (AR2)

**Response to EC**

Please, find our response to the EC comments below. Original comments are in black and our response is in blue.

\*\*

Dear authors and referees thank you for your contributions, the reviews and the revision of your manuscript.

Dear authors, your dataset and the well composed manuscript presents an interesting and relevant dataset on Vegetation Indices for the Iberian Peninsula and Balearic Islands. The reviewers state that the NDVI series are highly relevant for researchers, managers, land-owners, policy-makers as well as other stakeholders working on forestry, agriculture and other environmental sciences.

We are therefore pleased to accept your paper for publication in ESSD subject to minor revisions.

However currently the access to your datasets seem to be broken

https://doi.org/10.20350/digitalCSIC/16201 as well as

https://digital.csic.es/handle/10261/353068 seem not to be accessible

please provide a working link to the data in your manuscript.

In addition I would like to ask you to provide an additional usable technical readme file (reviewers 2 suggestion) in your data publication collection with standard technical information on the geodata publication (projection, resolution,...)

We ar looking forward to your final publication,

best regards, Birgit Heim

Dear Dr. Heim,

Thank you for your message and for accepting our paper subject to minor revisions.

Regarding the data access issue: the Digital.CSIC repository experienced a temporary service interruption, which caused the two links to be unreachable. The service has been restored and both links are now working:

https://doi.org/10.20350/digitalCSIC/16201

https://digital.csic.es/handle/10261/353068

In addition, following Reviewer 2's suggestion, we have added further technical information to the README file with key geospatial details (projection/CRS, resolution, file formats, variable metadata, temporal coverage, etc.). Please also note that the NetCDF files are self-describing, as they comply with the Climate and Forecast metadata conventions (CF conventions) and include the standard metadata required for data use.

We have also updated the reference list of the manuscript by adding DOIs wherever possible and by replacing the item previously "in review" with its published version.

Please let us know if anything else is needed.

Best regards,

Magí Franquesa (on behalf of all co-authors)